# Male and Female Perceptions of Supervision During Strength Training

**DOI:** 10.3390/sports12110301

**Published:** 2024-11-05

**Authors:** Luke Carlson, Maria Hauger, Grace Vaughan-Wenner, James P. Fisher

**Affiliations:** 1Discover Strength, 4450 Excelsior Blvd, Suite 490, St. Louis Park, MN 55416, USA; luke@discoverstrength.com (L.C.); maria@discoverstrength.com (M.H.); 2Discover Strength, 5041 France Ave S Suite 1, Edina, MN 55410, USA; grace@discoverstrength.com; 3Department of Sport and Health, Solent University, Southampton SO14 0YN, UK

**Keywords:** feedback, effort, technique, adherence

## Abstract

A cross-sectional survey was distributed to 1322 members of a 1-on-1 personalized strength training studio. A total of 366 respondents (*n* = 134 male and *n* = 232 female), all aged over 20 years, reported considerable training experience, with 55% of the males and 42% of the females reporting 5+ years of experience. The data were analyzed and reported descriptively with differences >5% identified based on the use of a 5-point Likert scale, the sample size, and the nature of the observations. Disparities between the males and females were identified; the males reported higher perceptions of managing effort, technique, and programming without supervision compared to the females. Safety was noted as being more important to the females compared to the males. Qualitatively, additional themes were raised including an analogy of the personal relationship between the trainer and trainee being similar to that between medical professionals and patients. This was validated where the participants discussed their adaptations from supervised strength training for maintaining quality of life in aging and recovering from medical conditions and injury. The data are discussed in the context of a previous body of literature suggesting males falsely report higher levels of confidence in tasks compared to females, particularly in relation to effort, role models, and verbal encouragement. We posit that the greater confidence expressed by males at least partially explains the greater engagement in strength training practices by males compared to females, as well as explaining the higher level of participation in supervised strength training by females compared to males. This research proves beneficial for strength training practitioners in enhancing their understanding and expectations of clients, as well as hopefully proving insightful in engaging more people in strength training.

## 1. Introduction

A recent systematic review and meta-analysis highlighted the importance of supervision during strength training for favorable physiological adaptations [1]. However, the authors also discussed a lack of research in the understanding of the role of supervision as well as trainer and trainee perceptions. In reviewing the literature, the authors highlighted the key characteristics of supervision to be technical coaching (correcting/maintaining client technique for exercise, providing feedback on performance), effort (increasing/maintaining sufficient effort level from the client to obtain desired adaptations), motivation (providing encouragement to complete a workout, to promote enjoyment, etc.), program design (designing training programs and workouts including exercise choice, load, progression, etc.), safety (spotting, technique correction to prevent injury, handing weights, etc.), and accountability (promoting adherence and engagement) [2]. In a subsequent survey of this topic completed by 468 trainers and trainees, the summary points were as follows: technical coaching was perceived to be the most important characteristic of supervision; persons engaging in supervised strength training reported lower injury rates compared to those training unsupervised; and females rated supervision more importantly than males [2]. Furthermore, qualitative comments from the participants suggested additional differences in males and females regarding the perceptions of supervision, and that male and female trainers might approach supervision to male and female trainees differently. However, since an investigation of sex differences in the perception to supervision was not the primary aim of this paper, the authors were hesitant to draw conclusions on this topic.

The participation rates for strength training remain overwhelmingly low, specifically for females (males = 18–35%, females = 14–26%) [3], with perceived complexity and difficulty often cited as barriers to engagement in resistance training [4]. However, while data suggest lower overall engagement in strength training, studies have reported higher engagement in supervised strength training by females compared to males from both trainers (~40% female, 17% male) and trainees (~55% female, ~19% male) [2]. While strength training has an array of health benefits for males and females [5,6], it might be particularly important throughout the lifespan in females for hormone regulation [7], the health of mothers and children during pregnancy and post-partum [8,9,10], and bone mineral density during and after menopause [11,12]. However, to date, there is no research which might explain the difference in engagement in supervised strength training.

In establishing a theoretical framework for this area, we believe that, based on the considerable experience of the authors as personal trainers and/or strength and conditioning coaches, there are important differences between males and females and their perceptions of supervision, which might determine engagement in strength training. Indeed, the need for more information on the key elements that form the basis of supervision in future research has been suggested to improve our understanding [1,13]. Furthermore, greater clarity on the differences in the perceptions of supervision between males and females during strength training might positively impact adherence and allow trainers and clients to strengthen their relationship by aligning their goals and expectations. As a product of knowledge enhancement resulting from this type of research, we might provide explanations as to the differences in engagement in strength training between males and females. With the above in mind, the aim of this study was to compare male and female perceptions of the characteristics of supervision during strength training.

## 2. Materials and Methods

### 2.1. Experimental Design

An online cross-sectional survey study was conducted between March and June 2024, with questions derived based on observations resulting from decades of combined experience working as personal trainers and/or strength and conditioning coaches. The overall structure of the questionnaire and questions themselves were piloted face to face with other trainers to gauge clarity and understanding. The primary research aim was to compare the perceptions of supervision between males and females, based on an exploratory rather than hypothesis-driven approach, but guided by previous research considering the characteristics of supervision. This study received ethical approval from the Health, Exercise, and Sport Science (HESS) Ethics Committee at Solent University on 1 March 2022 (fishj1HESS2022).

### 2.2. Sampling and Population

The survey was conducted in English and was distributed as a link via email to members of a 1-on-1 personalized strength training studio (Discover Strength, St. Louis Park, MN, USA). The email was sent to 1322 members, of which 60% were female. The survey was completed by 366 participants (~28% response rate). The first page of the survey provided participants with an information sheet containing all details of the study, and then confirmation they understood this and informed consent to participate were requested. Participant data is provided in Table 1.

### 2.3. Survey

The survey was administered through JISC Online Surveys (Bristol, UK). Following informed consent, general demographic details were required; age, biological sex assigned at birth, number of years engaging in strength training, and primary reason for engaging in strength training. Participants were then guided through a series of questions around their perceptions of supervision during strength training; the full survey can be found in the Appendix A. The survey asked about general perceptions of importance of each previously identified [2] characteristic of supervision (technical, effort, program design, motivational, accountability, and safety) using a 5-point Likert scale (not important, somewhat important, modestly important, very important, essential; see Table 2). Following this, the survey then asked 5 questions pertaining to the impact of supervision on each specific characteristic, once again using a 5-point Likert scale (strongly disagree, disagree, neither agree nor disagree, agree, strongly agree). These questions were worded to be similar in form between each characteristic (see Table 3, Table 4, Table 5, Table 6, Table 7 and Table 8 for questions). Finally, all participants were asked 9 additional questions about their perceptions of supervision for other characteristics (see Table 9) and were then invited to leave any qualitative comments should they wish to.

### 2.4. Statistical Analyses

The primary research aim was to compare perceptions of the importance of supervision between males and females. Data are reported by JISC software (version 2) as absolute and relative value responses by group (e.g., age, sex, etc.). From this, data have been reported as the most frequent responses as percentages of total respondents by group to accommodate variance between group values (females, *n* = 232; males, *n* = 134). Further analysis was performed using Microsoft Excel by grouping observation trends. For example, where a 5-point Likert scale (e.g., strongly disagree, disagree, neither agree nor disagree, agree, strongly agree) was used, data for “*strongly disagree*” and “*disagree*” were combined, and “*strongly agree*” and “*agree*” were combined. The same was applied for the scale “*not important*”, “*somewhat important*”, “*modestly important*”, “*very important*”, and “*essential*”. That is to say, “*not important*” and “*somewhat important*” were combined, as were “*very important*” and “*essential*”. This serves to reduce the data from an ordinal to a nominal level.

Data have been discussed where differences >5% were identified within and between groups at the nominal level, based on use of the 5-point Likert scale, sample size, and nature of the observations, although all data are available in the Appendix A. Ultimately, the >5% difference was chosen as a somewhat arbitrary cut-off deeming that a difference less than this suggested no discernible difference in response rate.

Qualitative content analysis was performed for questions that required an open text response [14]. Qualitative responses were assessed independently by three authors (James P. Fisher and Maria Hauger) and grouped and coded based on the supervision characteristics identified in the body of literature (i.e., technical, effort, motivation, program design, accountability, and safety), or any other categories consistently identified.

## 3. Results

### 3.1. Demographic Data

The survey was distributed to 1322 members of which ~60% were female. There were 366 responses, of which 134 were male (37%) and 232 were female (63%). All the participants were over the age of 20 years old, with 50% of the males and 45% of the females aged between 40 and 59 years, and 41% of the males and 39% of the females aged over 60 years. When asked the period of engagement in supervised resistance training, 10% of the males and 19% of the females reported < 1 year, while 55% of the males and 42% of the females reported 5+ years. In reporting the primary reason for engaging in strength training, 53% of the males and 47% of the females reported health improvements, while 31% of the males and 34% of the females reported physical function other than sports performance. The participants’ demographic data (age, sex, number of years training experience, and primary reason for engaging in resistance training) are presented in full in Table 1.

### 3.2. Quantitative Data

When asked to report their perception of the importance of each characteristic of supervision (Question 5, Table 2), the overriding responses were as follows: for technical coaching, 90% of the males and 97% of the females responded that it was very important or essential; for effort-based coaching, 94% of the males and 91% of the females responded that it was very important or essential; for program design, 90% of the males and 91% of the females responded that it was very important or essential; for motivation in the form of encouragement and enjoyment, 75% of the males and 85% of the females responded that it was very important or essential; for accountability, 79% of the males and 75% of the females responded that it was very important or essential; and for safety, 78% of the males and 92% of the females responded that it was very important or essential.

When asked a series of five specific questions relating to each characteristic (questions 6–11, Table 3, Table 4, Table 5, Table 6, Table 7 and Table 8), the overriding responses favored the importance of supervision. However, >5% differences based on the nominal values were identified as follows: Question 6.2: *My technique would always be good even without supervision*; 62% of the males vs. 70% of the females responded with “disagree/strongly disagree”. Question 7.2: *My effort level would always be high even without supervision*; 58% of the males vs. 69% of the females responded with “disagree/strongly disagree”. Question 7.4: *I dislike it when my PT pushes me too hard*; 81% of the males vs. 71% of the females responded with “disagree/strongly disagree”. Question 8.2: *I would feel confident to track and program my own workouts without a PT*; 58% of the males vs. 79% of the females responded with “disagree/strongly disagree”. Question 11.2: *My workouts would be equally safe even if I were not supervised a PT*; 69% of the males vs. 78% of the females responded with “disagree/strongly disagree”. Question 11.3: *I value the safety of having a PT present for my strength training workouts*; 87% of the males vs. 96% of the females responded with “agree/strongly agree”. Question 11.5: *The safety of my strength training is impactful in my ability to meet my goals*; 86% of the males vs. 93% of the females responded with “agree/strongly agree”.

Finally, nine uncategorized questions were asked about the impact of supervision upon other variables (Question 12; adaptations, efficiency, reducing anxiety, improving confidence, attitude towards strength training, feedback, confidence in personal trainer based on qualifications and experience, and enjoyment of working with different personal trainers). Once again, the data favored supervision for positive emotional perceptions (see Table 9). However, >5% differences based on the nominal values were identified as follows: Question 12.3: *Supervision by a PT reduces anxiety over performing strength training*; 72% of the males vs. 79% of the females responded with “agree/strongly agree”. Question 12.4: *Supervision by a PT improves confidence when performing strength training*; 90% of the males vs. 95% of the females responded with “agree/strongly agree”.

### 3.3. Qualitative Data

Question 13 invited the participants to “share any other thoughts around supervised strength training” using an open text box. Of the 366 participants who completed the survey, there were 120 (~33%; 37 male and 83 female) qualitative comments. Any comments which related solely to the survey itself were removed, leaving 97 responses (29 male and 68 female). These were then coded independently by three authors (JPF and MH). The previously identified characteristics were consistently mentioned with varying degrees of frequency. All the qualitative comments are available in the Appendix A (https://osf.io/rfxwb/); however, in brief, accountability was identified most frequently (10 comments by males and 24 by females; some comments included long-term accountability, and used terms such as consistency and frequency); program design occurred next most frequently (6 males and 21 females; comments often made reference to the knowledge/expertise of their trainer); motivation as a direct term was used by 2 males and 8 females; effort was identified 9 times (3 males and 6 females; comments included being “pushed to my limits” and “pushing past a comfort zone” by a personal trainer); technique occurred in 7 comments (all of which were from females; comments included the appreciation for “correcting technique”, “correcting my form”, and that “form is important” or that a trainer can “help me further improve my form”); finally, safety was identified 3 times (2 males and 1 female; as well as commenting on “feeling safe”, one comment was that the client feels “like I’m not going to injure myself with supervision”).

In addition to the previously identified characteristics, several other themes occurred with a degree of frequency. The first of these was personal relationships (including comments on professionalism and confidence; 9 males and 20 females). These comments encompassed the importance of “…[a] personal trainer that gets to know you and your goals”, and that “it’s important they learn me. My strengths. My weaknesses. My needs. My personality”, as well as the “attitude of the personal trainer”, and a “personal relationship with the trainer is a significant factor”. Further terms including feeling cared for, or trainers being friendly. With regard to professionalism, the clients stated, “The trainers have a superb work ethic and professionalism in addition to being extremely knowledgeable” and “The trainers are great. Professional, friendly and really motivating”. One male and one female participant both used the term “confidence”, stating, “Having a trainer brings a level of confidence when working out” and “Work[ing] with a trainer on a consistent basis builds my trust and confidence”, respectively. These subthemes (professionalism and confidence) were grouped together because it was felt they were indicative of the trainer themselves, their attitude, or their approach to a client. A number of comments in the same theme identified that they did not like switching between trainers because they value the personal relationship: “I think it’s really important for the trainer to know you”, “I like the continuity of having the same trainer rather than, as is often the case, getting a rotation of different and new trainers”, “Having too many trainers to have to work with is difficult as I need to train them on how to train me”, and “Benefits can be very trainer specific … some trainers don’t realize that different clients need to be motivated/pushed to different extents—one size does not fit all”. The theme of personal relationships and professionalism is perhaps embodied best when a participant commented “I have become very attached to certain trainers just like a dentist or doctor.”

The final theme identified was related to adaptations resulting from strength training (4 males and 19 females). Many participants referred to the benefits of strength training with respect to aging: “…the value it [strength training] has for me mentally and physically today but also as I age”. Other comments identified the benefits of combatting ill-health: “Strength training kept me strong during 9 months of cancer treatment now 5 years ago, and I continue to make progress in building strength at age 68”, and “I gained even more appreciation for supervised training after my 2022 stroke”. Also, recovery from injury was mentioned: “I have healed multiple injuries through strength training with supervision”. Finally, quality of life was discussed: “As a 75-year-old male, I strongly believe that without regular strength training, my quality of life would not be what it is today”.

## 4. Discussion

The results from the present study highlight the disparities and similarities in the perceptions of supervision, and the characteristics thereof, during strength training between males and females. In general, all the data supported favorable responses towards supervision during strength training. This is as expected by a group of people presently paying for supervised strength training. While it is fair to argue that the population group considered see the importance of supervision and so engage in it, there is also the possibly of a confirmation bias, i.e., that because they pay for supervised strength training, they are more likely to report favorably for its importance. It is also worth highlighting the participant demographics, as 45–50% of the males and females were aged between 40 and 59 years, and ~40% of the males and females were aged over 60 years. The seniority of this demographic supports the possibility of greater training experience (e.g., 42–55% of the participants had >5 years), as well as the primary reasons for engaging in strength training; ~50% of the males and females reported “*health improvements*”, while ~30% of the males and females reported “*physical function other than sports performance*”. Finally, it is perhaps noteworthy that this demographic is more likely to have a disposable income and can afford supervised strength training.

The initial questions which asked simply about the importance of each characteristic of supervision produced some interesting data. The between-group comparisons of the responses suggest that males place less importance on supervision for technical coaching, motivation for enjoyment, and safety compared to females. A disparity in the importance of these characteristics was also identifiable in the subsequent questions on each specific characteristic. For example, when responding to “*My technique would always be good even without supervision*”, a greater number of females compared to males disagreed or strongly disagreed. Presented another way, a larger number of males seemed more confident that they could maintain good technique without supervision. It is worth recognizing that the dominant response from the males still favored that they disagreed that they could not maintain good technique (i.e., only 17% of the males agreed with the statement), but there was a lower percentage of participants giving this response (see Table 3). Further, when answering “*I would feel confident to track and program my own workouts without a PT*”, a larger number of females compared to males disagreed. Thus, a larger number of males compared to females (19% vs. 6%, respectively) agreed that they could track and program their own workouts (see Table 5). Finally, when responding to “*My effort level would always be high even without supervision*”, the males typically reported higher agreement (and lower disagreement) compared to the females. This suggests that a greater number of male respondents perceive that they can maintain a similarly high effort level without supervision. In support, when responding to “*I dislike it when my PT pushes me too hard*”, a greater number of males disagreed with this statement compared to females. We deliberately chose the phrasing “*too hard*” as something that all could (or should) identify as an extreme beyond an acceptable range (e.g., *too* hot, or *too* cold). However, the data suggest that females might be more likely not to want to work outside of extreme effort levels, whereas males perceive effort as more important and manageable independently as well as in extremes (see Table 4).

This confidence might relate to the larger number of males reporting greater strength training experience (e.g., 65% of the males compared to 50% of the females had 3 or more years of training experience; see Table 1). However, a body of literature suggests a disparity in self-confidence between the male and females. It is well documented that males have higher self-confidence levels compared to females in various parameters including education [15], employment [16], and physical activity and motor control [17]. Indeed, Dweck et al. [18,19] discuss that this confidence might arise from school years, where positive feedback in the classroom might encourage females to be enthused to attain this praise and less likely to take risks and make mistakes, weakening their confidence. In contrast, males of the same age (who typically have lower attention spans, less advanced verbal and fine motor skills, and lesser social adeptness) typically receive feedback based on effort (or a lack thereof). This might fuel them to perceive that performance in a task is not outside their ability only dependent upon their application—something echoed herein in responses to effort, programming, and exercise technique. In a further study, reviewing educational pursuits, the authors reported that males are often more influenced by positive male role models, whereas females appeared to benefit from verbal encouragement and contextualization [20]. In an exercise environment, males might present confidence in their abilities based on their experience of males depicted in social media, or other forms of role models, whereas females might be less confident in general but also attain confidence through verbal encouragement from personal trainers themselves.

Further, Corbin et al. [21] suggested that the confidence disparity between males and females was not trait- but state-specific. They continued by using the term “*situationally vulnerable*”, i.e., lacking confidence in specific environments. Certainly, the exercise industry might be historically thought of as male-dominated; a meta-analysis considering self-confidence in physical activity reported an effect size of 0.40 in favor of greater self-confidence in males compared to females [22]. Even in younger children, perceptions of tasks requiring strength, speed, and power are characterized as male activities (by both males and females) with confidence in successful performance higher in males compared to females [17]. Furthermore, our data showing males having greater confidence in their capacity to train without supervision are supported by a previous survey, where people participating in regular strength training suggested that a primary reason for non-engagement in supervision was a perception that their knowledge and experience was sufficient not to warrant a trainer [1]. In addition, data have previously reported a near 27:1 ratio of males to females in a free-weight section of a gym [23], which, while potentially indicative of low participation in strength training by females, might also be indicative of a lack of knowledge and/or confidence in strength training with free weights.

In contrast to males reporting greater importance and possibly autonomy in effort, technique, and program design, males typically seemed not to rate safety with the same importance as females. A greater number of females (i) disagreed that their workouts would be equally safe without supervision, (ii) agreed that they valued the safety of having a personal trainer present for their workouts, and (iii) agreed that the safety of their strength training is impactful in their ability to meet their goals (Table 8) compared to males. However, this was due to a larger number of males responding “*neither agree nor disagree*” rather than answering with an opposing view. That females feel safer with supervision compared to males links to the above narrative, highlighting a disparity in confidence between males and females. That is to say, males might feel sufficiently confident engaging in strength training such that supervision does not enhance their feeling of safety.

Our final group of questions were exploratory in nature rather than themed around the previously identified characteristics. While some of these questions were very general in nature (e.g., “*Supervision by a personal trainer optimizes strength training adaptations*”), others posed questions relating to the attitude towards strength training, which might provide insight toward promoting greater adherence. For example, when responding to “*Supervision by a personal trainer reduces anxiety over performing strength training*” and “*Supervision by a PT improves confidence when performing strength training*”, a greater number of females agreed with these statements compared to males. However, once again, the proportion of males which did not agree did not answer in the contrary but rather answered “*neither agree nor disagree*”, suggesting that they do not present opposing views but simply do not feel as strongly. These responses add further support for the previous theme of male confidence, specifically in strength training. Notably, some of the questions related specifically to clients of Discover Strength; for example, all the personal trainers are required to have a bachelor’s degree and be at the least American College of Sports Medicine-certified as an exercise physiologist. In addition, clients do not book with a trainer per se, but rather book a time and are assigned a trainer working that shift. Based on these points, questions such as “*I have confidence in my personal trainer because of their qualifications*”, and “*I enjoy working with a team of different personal trainers*” were deemed important but did not produce differing responses between the males and females.

Finally, it is worth noting the motivation displayed by the current cohort of participants. While we recognize that many of them have been engaging in 1-on-1 personalized strength training for a long period, it is also worth noting that the section asking about adherence (e.g., “*My strength training workouts are more frequent because of supervision by a PT*”; Table 7) received the lowest positive score compared to the other characteristics. It seems that, within the present participants, accountability/adherence, while important, is of equally lesser importance to males and females. From this perspective, it is perhaps worth reviewing accountability as a characteristic of supervision. For example, while supervision might promote greater adherence through accountability [1], there are studies where adherence was similar in the supervised and unsupervised conditions [6,24,25], and still, physical outcome measures favored supervised training conditions.

In contrast, the analysis of the qualitative data identified accountability/adherence as the most frequently discussed characteristic (*n* = 37). These included comments around long-term adherence and maintaining consistency being more achievable with supervision compared to previous experiences of training alone. Program design was also mentioned with a degree of frequency (*n* = 27), including comments of knowledge and expertise. While these comments might be specific to the trainers at Discover Strength as an organization based on the minimum required qualifications, it is worth recognizing the importance of knowledge/expertise in exercise professionals and the confidence this bestows in clients, since a lack of knowledge/perceived complexity is an often-cited barrier to strength training [4]. While these qualitative comments might appear contradictory to the discussion of confidence to program a workout and train without supervision, we should recognize that the majority of the participants responded favorably to supervision for program design and that these qualitative comments make up only a small percentage (~7%) of the total participants.

The qualitative data also identified the themes of professional relationships (also using the terms “confidence” and “professionalism”) and adaptations. Much of the previous literature appears to have overlooked the importance of the personal relationship between the trainer and trainee. However, this was stressed with many positive comments as well as some participants responding that they valued consistency and did not like working with different trainers. This theme was perhaps clarified best when a participant made the analogy of their personal trainer to other medical professionals—that of a dentist or doctor. This comparison was validated when other participants responded by highlighting the positive adaptations from strength training for quality of life as they age, as well as for their health during medical conditions (cancer treatment, stroke, recovery from injury). All qualitative data are available in the Appendix A.

### Limitations

We should acknowledge the possible limitations of the present study. While there appears to be an apparent self-confidence in males above females in their capacity to display good technique, exert a sufficiently high effort, and program and track their workouts, and this is consistent with previous literature, we should recognize some key points. Firstly, previous studies which have discussed this (over)confidence portrayed by males have then demonstrated that it was unjustified, and that, when assessed, they did not perform better than females in the given tasks. In this instance, we cannot know whether males would be better in their technical and effort performance during strength training, or their capacity to program and track their workouts, only that they believe so. Secondly, with a growing body of social media sites discussing and portraying strength training and strength training guidance, it might be reasonable that males view this media more so than females, which has fueled a perception that they might be more capable on the points raised. A further possible limitation is the age and training experience of the participants in this study. While the use of a personalized strength training studio to contact clients to complete this survey ensures that all the respondents were engaging in personal, supervised strength training, it might also limit the extent to which our data can be extrapolated to a wider population. Certainly, we cannot assume that previously untrained participants beginning strength training would place importance on the same characteristics to the same extent, or that they might share similar perceptions about their capacity to perform strength training without supervision.

## 5. Conclusions

In summary, our data suggest that males place greater importance on supervision for technical coaching, and less importance on safety and enjoyment as characteristics of supervision during strength training, compared to females. However, the data also suggest that males are confident that they are more capable of tracking and programming their own workouts and maintaining sufficiently high effort and quality of technique. This is supported by a body of literature considering higher self-confidence in males compared to females, potentially in particular relation to effort, role models, and verbal encouragement. We believe this combined body of research can ultimately add to the theoretical framework underpinning our understanding of male and female participation in strength training. Further, where a lack of knowledge/perceived complexity are often cited barriers to engagement and adherence to strength training [4], it seems likely that these factors are more influential over females compared to males. The present study improves our understanding of the confidence in males, and lack thereof in females, and we posit that this plays a role in the greater engagement in strength training practices by males compared to females, as well as explaining the higher participation in supervised strength training by females compared to males. Thus, while supervision can serve to enhance trainer–trainee relationships, this appears to be of greater importance to females.

All the participants reported favorably regarding their physiology and adaptation to strength training, including an analogy to medical professionals (doctors and dentists). Based on the overwhelming body of literature supporting the health benefits of strength training, this comparison seems justifiable and personal trainers and strength coaches should consider the importance and positive health implications when forging a trainer–trainee relationship. Finally, we encourage strength coaches and personal trainers to engage in the findings of this research to strengthen their understanding of client motives and sex differences, placing particular importance upon the motives of females who are underrepresented as engaging in strength training.

## Figures and Tables

**Table 1 sports-12-00301-t001:** Participant demographics.

Question		Male (*n* = 134; 36.6%)	Female (*n* = 232; 63.4%)
Age (years)	20–29	4 (3%)	9 (4%)
	30–39	9 (7%)	27 (12%)
	40–49	41 (31%)	41 (18%)
	50–59	25 (19%)	64 (28%)
	60–69	41 (31%)	63 (27%)
	70+	14 (10%)	28 (12%)
Training experience (years)	<1	13 (10%)	43 (19%)
	1–2	18 (13%)	16 (16%)
	2–3	16 (12%)	34 (15%)
	3–4	14 (10%)	19 (8%)
	5+	73 (55%)	120 (42%)
Primary reason for engaging in strength training	Sporting/athletic performance	8 (6%)	10 (4%)
	Health improvements	71 (53%)	109 (47%)
	Ill health avoidance	11 (8%)	33 (14%)
	Social reasons	0%	0%
	Aesthetics	2 (2%)	1 (<1%)
	Physical function	42 (31%)	79 (34%)

**Table 2 sports-12-00301-t002:** Question 5. Please rate the importance of supervision for these characteristics of strength training.

	Response									
Characteristic	Essential		Very Important		Modestly Important		Somewhat Important		Not important	
	Male	Female	Male	Female	Male	Female	Male	Female	Male	Female
Technical—Correcting/maintaining your form	50.0%	74.6%	39.6%	22.0%	10.4%	2.6%	0.0%	0.9%	0.0%	0.0%
Effort—Increasing/maintaining your effort level	61.9%	59.9%	32.1%	31.5%	4.5%	6.0%	1.5%	2.2%	0.0%	0.4%
Program Design—Exercise choice, load, progression, etc	40.3%	55.6%	46.3%	35.3%	11.9%	8.2%	1.5%	0.9%	0.0%	0.0%
Motivational—Providing encouragement to complete a workout	36.4%	39.2%	39.4%	45.7%	19.7%	10.3%	4.5%	2.6%	1.5%	2.2%
Accountability—Promoting engagement and adherence	36.8%	40.1%	42.9%	34.5%	12.8%	19.4%	7.5%	3.4%	0.8%	2.6%
Safety—Spotting, technique correction to prevent injury, handing weights, etc.	48.5%	71.6%	30.3%	20.7%	12.1%	6.0%	9.1%	1.3%	1.5%	0.4%

All values reported as percentage of total group (female = 232, male = 134).

**Table 3 sports-12-00301-t003:** Question 6. Please rate the importance of supervision for technique during strength training.

	Response									
Statement	Strongly Agree		Agree		Neither Agree nor Disagree		Disagree		Strongly Disagree	
	Female	Male	Female	Male	Female	Male	Female	Male	Female	Male
My technique is good because my strength training workouts are supervised	73.7	59.7	22.0	35.8	3.0	4.5	0.9	0.0	0.4	0.0
My technique would always be good even without supervision	2.6	1.5	9.5	15.7	18.1	20.9	53.0	48.5	16.8	13.4
I value feedback about my technique from a personal trainer	79.7	70.1	19.4	29.1	0.9	0.7	0.0	0.0	0.0	0.0
I dislike it when my personal trainer corrects my technique	1.3	0.0	1.3	0.7	1.3	5.2	19.4	27.6	76.7	66.4
Technique during strength training is impactful in my ability to meet my desired goals	67.7	54.5	28.9	39.6	3.0	3.0	0.4	2.2	0.0	0.7

All values reported as percentage of total group (female = 232, male = 134).

**Table 4 sports-12-00301-t004:** Question 7. Please rate the importance of supervision for effort during strength training.

	Response									
Statement	Strongly Agree		Agree		Neither Agree nor Disagree		Disagree		Strongly Disagree	
	Female	Male	Female	Male	Female	Male	Female	Male	Female	Male
My effort level is highest because my strength training workouts are supervised	70.3	64.2	24.6	30.6	3.0	3.0	1.7	1.5	0.4	0.7
My effort level would always be high even without supervision	2.2	3.0	15.1	19.4	14.2	19.4	50.0	44.8	18.5	13.4
I value a personal trainer pushing me to work harder	70.3	68.7	24.6	26.9	3.0	3.7	0.4	0.7	1.7	0.0
I dislike it when my personal trainer pushes me too hard	3.9	0.0	12.1	7.5	13.4	11.2	30.2	33.6	40.5	47.8
Effort during strength training is impactful in my ability to meet my desired goals	72.8	73.1	25.0	26.1	0.9	0.7	0.9	0.0	0.4	0.0

All values reported as percentage of total group (female = 232, male = 134).

**Table 5 sports-12-00301-t005:** Question 8. Please rate the importance of program design during strength training.

	Response									
Statement	Strongly Agree		Agree		Neither Agree nor Disagree		Disagree		Strongly Disagree	
	Female	Male	Female	Male	Female	Male	Female	Male	Female	Male
My strength training workouts are better because of programming by a personal trainer	79.3	69.4	18.1	27.6	2.6	3.0	0.0	0.0	0.0	0.0
I would feel confident to track and program my own workouts without a personal trainer	1.3	5.2	5.2	14.2	14.2	22.4	43.1	44.0	36.2	14.2
I value my personal trainer tracking and progressing my workouts over time	68.5	59.0	30.2	36.6	0.9	4.5	0.4	0.0	0.0	0.0
I dislike it when my personal trainer progresses my load/changes exercises without discussion	2.2	1.5	6.0	8.2	13.8	14.2	40.9	36.6	37.1	39.6
The strength training program I use is impactful in my ability to meet my goals	68.5	69.4	28.4	29.9	2.2	0.7	0.4	0.0	0.4	0.0

All values reported as percentage of total group (female = 232, male = 134).

**Table 6 sports-12-00301-t006:** Question 9. Please rate the importance of motivation during strength training.

	Response									
Statement	Strongly Agree		Agree		Neither Agree nor Disagree		Disagree		Strongly Disagree	
	Female	Male	Female	Male	Female	Male	Female	Male	Female	Male
My strength training workouts are more enjoyable because of supervision by a personal trainer	68.1	62.7	25.4	32.1	6.5	5.2	0.0	0.0	0.0	0.0
I would enjoy my workouts to a similar extent without supervision	1.3	0.7	8.6	8.2	11.6	17.2	47.8	56.7	30.6	17.2
I value a personal trainer encouraging me with positive feedback	69.0	59.0	28.4	38.8	2.2	2.2	0.0	0.0	0.4	0.0
I dislike it when my personal trainer tries to motivate me	1.7	0.0	2.6	0.0	3.4	6.0	37.5	38.1	54.7	56.0
My enjoyment of strength training is impactful in my ability to meet my goals	58.2	56.7	34.5	35.8	6.5	6.7	0.9	0.7	0.0	0.0

All values reported as percentage of total group (female = 232, male = 134).

**Table 7 sports-12-00301-t007:** Question 10. Please rate the importance of accountability during strength training.

	Response									
Statement	Strongly Agree		Agree		Neither Agree nor Disagree		Disagree		Strongly Disagree	
	Female	Male	Female	Male	Female	Male	Female	Male	Female	Male
My strength training workouts are more frequent because of supervision by a personal trainer	49.1	41.8	26.3	31.3	17.7	18.7	5.6	7.5	1.3	0.7
My workouts would have similar frequency if I were not meeting a personal trainer	3.4	3.0	9.5	11.9	12.1	14.2	41.8	47.8	33.2	23.1
I value the accountability of meeting personal trainer for strength training workouts	67.2	60.4	28.9	32.8	3.4	6.7	0.0	0.0	0.4	0.0
I dislike having to schedule strength training workouts	1.3	0.0	4.3	6.0	7.8	10.4	40.9	45.5	45.7	38.1
The regularity of my strength training is impactful in my ability to meet my goals	78.0	73.9	20.3	26.1	0.9	0.0	0.4	0.0	0.4	0.0

All values reported as percentage of total group (female = 232, male = 134).

**Table 8 sports-12-00301-t008:** Question 11. Please rate the importance of safety during strength training.

	Response									
Statement	Strongly Agree		Agree		Neither Agree nor Disagree		Disagree		Strongly Disagree	
	Female	Male	Female	Male	Female	Male	Female	Male	Female	Male
My strength training workouts are safer because of supervision by a personal trainer	75.4	53.7	22.0	39.6	1.3	6.7	1.3	0.0	0.0	0.0
My workouts would be equally safe even if I were not supervised a personal trainer	1.3	0.7	6.0	9.7	14.2	20.9	44.4	47.8	34.1	20.9
I value the safety of having a personal trainer present for my strength training workouts	67.2	48.5	28.9	38.8	3.0	11.2	0.4	1.5	0.4	0.0
My strength training workouts are too focused on safety	1.3	0.0	0.4	1.5	6.9	4.5	41.8	53.7	49.6	40.3
The safety of my strength training is impactful in my ability to meet my goals	63.4	45.5	29.3	40.3	6.0	10.4	1.3	3.7	0.0	0.0

All values reported as percentage of total group (female = 232, male = 134).

**Table 9 sports-12-00301-t009:** Question 12. Other questions relating to perceptions of supervision.

	Response									
Statement	Strongly Agree		Agree		Neither Agree nor Disagree		Disagree		Strongly Disagree	
	Female	Male	Female	Male	Female	Male	Female	Male	Female	Male
Supervision by a PT optimizes strength training adaptations	67.7	59.0	28.9	32.8	3.4	7.5	0.0	0.7	0.0	0.0
Supervision by a PT optimizes efficiency of a strength training workout	76.3	66.4	22.0	32.1	1.7	1.5	0.0	0.0	0.0	0.0
Supervision by a PT reduces anxiety over performing strength training	46.1	32.1	33.2	40.3	12.9	21.6	5.6	5.2	2.2	0.7
Supervision by a PT improves confidence when performing strength training	63.4	48.5	31.9	41.8	3.9	8.2	0.4	0.7	0.4	0.7
I have a more positive attitude toward strength training because of supervision by a PT	65.5	58.2	23.7	28.4	7.8	11.9	3.0	0.7	0.0	0.7
Feedback from a PT is an important part of my strength training	63.4	64.9	31.0	31.3	4.7	3.7	0.9	0.0	0.0	0.0
I have confidence in my PT because of their experience	70.3	62.7	26.3	33.6	3.0	3.0	0.4	0.7	0.0	0.0
I have confidence in my PT because of their qualifications	63.8	47.8	31.9	46.3	3.4	6.0	0.9	0.0	0.0	0.0
I enjoy working with a team of different PTs	31.0	32.8	34.1	36.6	22.4	17.9	9.5	11.9	3.0	0.7

(PT = personal trainer).

## Data Availability

All survey questions, quantitative and qualitative data, and analyses are available in the Appendix A.

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
