# Peer review of "Male and Female Perceptions of Supervision During Strength Training"

_sports, 2024, doi:10.3390/sports12110301_

Round 1

Reviewer 1 Report

Comments and Suggestions for Authors

A survey of 366 members from a personalized strength training studio revealed gender differences in training perceptions and priorities. Males reported higher confidence in managing effort, technique, and programming independently, while females placed a greater emphasis on safety. Qualitative data showed that participants often compared the trainer-trainee relationship to that with medical professionals, with reported benefits such as improved quality of life and recovery from injuries. These findings highlight the importance of tailored approaches in strength training to better meet the unique expectations and needs of clients, potentially encouraging wider engagement in strength training.

General Comments:

The manuscript was clear, relevant for the field, and was presented in a well-structured manner.

The cited references were relevant and had both recent and older publications (within the last 5 years). There were not an excessive number of self-citations.

The authors provided sufficient detail in the methods section for reproducibility.

The figures/tables/images/schemes are appropriate and properly show the data. The figures are easy to interpret and understand. The data appears to be interpreted appropriately and consistently throughout the manuscript.

The authors’ conclusions were consistent with the evidence and arguments presented.

This and future studies on this topic could help fill a gap in knowledge.

Specific Comments:

Line #31: Provide clarification for “area”

Line #43: The last sentence in this paragraph needs to be rephrased for clarity

Line #50: I like the encompassing benefits of strength training for females, but you would need to provide some additional context, for example, why for “pregnancy?”

Line #56: How would you know there is a lack of transparency?

Line #61: Consider adding a comma after “mind”

Line #62: Consider adding context to “perceptions of supervision” as this seems broad and vague

Line #71: Detail on the location of the gym would be favorable, so readers can assess for any geographical/cultural/social implications

Line #94: Was any statistical software used?

Line #112: Consistency of decimal points would be favorable..”50.0%..45%...”

Line #126: Consider using another way of reporting in the paragraph without using “=” signs

Line #146+: Commas are typically placed within the quotation marks, e.g. “…strength training,”…

Line #174: formatting appears off

Line #235: Is bolding of “too hard” necessary?

Line #241-275: This is fairly long paragraph, is it possible to divide it?

Line #352: Not sure if it is grammatically correct to have i.e. as a standalone in a sentence

Like #356: “cites” should be spelled “sites”

References: Look to be appropriate and properly structured.

Author Response

Reviewer 1

Responses

The manuscript was clear, relevant for the field, and was presented in a well-structured manner.

The cited references were relevant and had both recent and older publications (within the last 5 years). There were not an excessive number of self-citations.

The authors provided sufficient detail in the methods section for reproducibility.

The figures/tables/images/schemes are appropriate and properly show the data. The figures are easy to interpret and understand. The data appears to be interpreted appropriately and consistently throughout the manuscript.

The authors’ conclusions were consistent with the evidence and arguments presented.

This and future studies on this topic could help fill a gap in knowledge.

We thank the reviewer for their detailed review and positive and constructive comments to improve the quality of this manuscript.

Line #31: Provide clarification for “area”

This has been amended to “literature”

Line #43: The last sentence in this paragraph needs to be rephrased for clarity

Agreed, this has been amended.

Line #50: I like the encompassing benefits of strength training for females, but you would need to provide some additional context, for example, why for “pregnancy?”

Agreed, this has been expanded

Line #56: How would you know there is a lack of transparency?

Agreed, this has been amended.

Line #61: Consider adding a comma after “mind”

Added.

Line #62: Consider adding context to “perceptions of supervision” as this seems broad and vague

Agreed, this has been expanded.

Line #71: Detail on the location of the gym would be favorable, so readers can assess for any geographical/cultural/social implications

Agreed, we have added that in this instance it was Discover Strength, Minnesota, USA. Although for clarity Discover Strength also operates a number of locations in other states as well as franchise locations.

Line #94: Was any statistical software used?

The statistical analysis section has been extensively rewritten based on the comments of reviewer two. Discussion of additional software has been included.

Line #112: Consistency of decimal points would be favorable..”50.0%..45%...”

Agreed, and amended for consistency

Line #126: Consider using another way of reporting in the paragraph without using “=” signs

Agreed, the “=” signs have been removed

Line #146+: Commas are typically placed within the quotation marks, e.g. “…strength training,”…

The comma has now been removed

Line #174: formatting appears off

Amended.

Line #235: Is bolding of “too hard” necessary?

Amended.

Line #241-275: This is fairly long paragraph, is it possible to divide it?

Agreed, this paragraph has been divided appropriately.

Line #352: Not sure if it is grammatically correct to have i.e. as a standalone in a sentence

This has been amended accordingly.

Like #356: “cites” should be spelled “sites”

Amended.

References: Look to be appropriate and properly structured.

Thank you.

We would like the reviewer to note that their suggestions were adhered to (see above) however, some of these changes might not be immediately identifiable because of the significant changes to the manuscript based on the comments from reviewer 2.

Reviewer 2 Report

Comments and Suggestions for Authors

Overview

The study reports on a survey of 366 strength training participants. Results revealed that males reported higher confidence in managing effort and technique, while females prioritized safety. There are some strengths:

a)  Large data set: The study utilizes a substantial sample size of 366 respondents

b) Applied research question: The research addresses practical, real-world issues related to strength training, making it highly relevant and applicable to both practitioners and participants.

c) Consideration of findings' translation to practice: The study thoughtfully considers how its findings can be implemented to improve practitioner understanding and client engagement, enhancing its practical value.

There are some limitations. 

Scientific protocol: The study lacks a clear adherence to scientific research standards, particularly in its methodology, structure, and reporting. To improve, the paper needs a more systematic approach, including clearly defined research objectives, detailed methodological explanations, and transparent data analysis. This would ensure replicability and enhance its scientific credibility, aligning with accepted research practices.

A weak literature review: The literature review does not sufficiently frame the research in the context of existing work. It needs to provide a stronger rationale by reviewing relevant studies that highlight the knowledge gaps the current research aims to fill. A robust literature review would not only build the case for the study but also contextualize the findings.

A Lack of theoretical framework: The study could be enhanced by integrating theories related to confidence, gender differences in training, or strength training behavior. A theoretical framework would provide a foundation for interpreting the results and linking them to broader psychological or physiological principles, offering deeper insight and increasing the study’s academic and practical value.

Methodological justification: The chosen methods are not adequately justified, leaving readers unclear about why specific approaches were used. The authors need to explain why these methods, such as the survey design and data analysis techniques, were suitable for addressing the research question. Providing a rationale would strengthen the methodological rigor and the trustworthiness of the conclusions. Related to this point, is questionnaire development: The development of the custom questionnaire requires further explanation, particularly regarding how the questions were derived, tested for reliability, and validated for the target population. Without this information, the credibility of the findings may be questioned. Clear articulation of the design process would provide confidence in the tools used and ensure the results are reliable.

The authors should consider whether the participants who completed the research are representative of the broader population from which they were drawn. The sample is predominantly individuals with over 5 years of training experience and aged over 40, which may limit the generalizability of the findings. This specific demographic may not reflect the diversity of strength training participants, particularly those newer to training or from younger age groups. Addressing the potential bias in participant selection and discussing its impact on the study's conclusions would help to strengthen the validity of the results and their applicability to a wider audience.  

The work would benefit from a clearly defined and thoroughly explained data analysis strategy. The current approach lacks detail on how the data were processed, analyzed, and interpreted. A more explicit description of the statistical techniques or qualitative analysis methods used would enhance the transparency and rigor of the study. For instance, outlining how differences between male and female participants were identified, and the rationale behind choosing specific thresholds (e.g., the 5% difference) would provide clarity. Furthermore, justifying the use of descriptive statistics and any inferential methods would help readers understand how the authors arrived at their conclusions, thereby strengthening the overall credibility of the findings.

The revisions require a considerable revision.

Comments on the Quality of English Language

The English is fine

Author Response

Reviewer 2

Responses

The study reports on a survey of 366 strength training participants. Results revealed that males reported higher confidence in managing effort and technique, while females prioritized safety. There are some strengths:

a)  Large data set: The study utilizes a substantial sample size of 366 respondents

b) Applied research question: The research addresses practical, real-world issues related to strength training, making it highly relevant and applicable to both practitioners and participants.

c) Consideration of findings' translation to practice: The study thoughtfully considers how its findings can be implemented to improve practitioner understanding and client engagement, enhancing its practical value.

We appreciate the reviewer taking their time to consider this manuscript and the positive comments provided.

There are some limitations.

Scientific protocol: The study lacks a clear adherence to scientific research standards, particularly in its methodology, structure, and reporting. To improve, the paper needs a more systematic approach, including clearly defined research objectives, detailed methodological explanations, and transparent data analysis. This would ensure replicability and enhance its scientific credibility, aligning with accepted research practices.

We thank the reviewer for this comment and appreciate that it might appear to lack adherence to research standards. In an effort to improve the quality of the manuscript we have added detail to the Experimental Design section for the reader to better understanding the research objectives. We have also added additional detail to the other Methods sections.

The data analysis is purely descriptive (e.g., the percentage of responses dominantly favoured). For example, when asked “my technique is good because my strength training workouts are supervised” we have provided data which showed 96% of males and females responded agree or strongly agree. In the instance of a likert scale it would be foolish to think that if 90% of males reported agree and 90% of females reported strongly agree that there is a difference. Further, where an equally small percentage of people might have reported disagree or strongly disagree, it doesn’t seem necessary to publish all data. Although this is available at the links provided.

A weak literature review: The literature review does not sufficiently frame the research in the context of existing work. It needs to provide a stronger rationale by reviewing relevant studies that highlight the knowledge gaps the current research aims to fill. A robust literature review would not only build the case for the study but also contextualize the findings.

We fully appreciate the reviewer’s comments, however, we are limited by a relative dearth of literature discussing the area of supervision during strength training. The reality is that there are so few studies that only those cited have even identified these knowledge gaps, and we believe we have clarified the gaps we aimed to fill. Since there is a lack of research to discuss and since this is an exploratory study, we aimed to keep the introduction brief. That said, we have added detail to the introduction in light of this comment to hopefully provide clarity to the reader.

A Lack of theoretical framework: The study could be enhanced by integrating theories related to confidence, gender differences in training, or strength training behavior. A theoretical framework would provide a foundation for interpreting the results and linking them to broader psychological or physiological principles, offering deeper insight and increasing the study’s academic and practical value.

Many thanks for this comment, we have added extensively to our introduction and methods as a result of this, as well as our conclusion. Ultimately, the confidence by male’s likely links to the higher engagement in strength training practices by males.

Methodological justification: The chosen methods are not adequately justified, leaving readers unclear about why specific approaches were used. The authors need to explain why these methods, such as the survey design and data analysis techniques, were suitable for addressing the research question. Providing a rationale would strengthen the methodological rigor and the trustworthiness of the conclusions. Related to this point, is questionnaire development: The development of the custom questionnaire requires further explanation, particularly regarding how the questions were derived, tested for reliability, and validated for the target population. Without this information, the credibility of the findings may be questioned. Clear articulation of the design process would provide confidence in the tools used and ensure the results are reliable.

We appreciate this comment and have added some detail to support our methods. However, we also believe that, as an exploratory piece of research, the use of a survey (as an established research method) is suitable. The questions were derived based on considerable experience of the authors as personal trainers and/or strength and conditioning coaches and the survey was piloted with other coaches to assess for clarity and understanding. We don’t believe this is the most groundbreaking piece of research but as an exploratory piece of research at this stage in establishing a more rigorous theoretical framework we felt this was appropriate. Furthermore, the final question was to ask participants to leave personal details should they be willing to be contacted for a follow-up semi-structured interview (underway) which might present a more rigorous approach to this research topic.

The authors should consider whether the participants who completed the research are representative of the broader population from which they were drawn. The sample is predominantly individuals with over 5 years of training experience and aged over 40, which may limit the generalizability of the findings. This specific demographic may not reflect the diversity of strength training participants, particularly those newer to training or from younger age groups. Addressing the potential bias in participant selection and discussing its impact on the study's conclusions would help to strengthen the validity of the results and their applicability to a wider audience. 

We thank the reviewer for this comment and while we had attempted to clarify this limitation in the appropriate sections (limitations) in the discussion, we have now added more detail and clarity.

The work would benefit from a clearly defined and thoroughly explained data analysis strategy. The current approach lacks detail on how the data were processed, analyzed, and interpreted. A more explicit description of the statistical techniques or qualitative analysis methods used would enhance the transparency and rigor of the study. For instance, outlining how differences between male and female participants were identified, and the rationale behind choosing specific thresholds (e.g., the 5% difference) would provide clarity. Furthermore, justifying the use of descriptive statistics and any inferential methods would help readers understand how the authors arrived at their conclusions, thereby strengthening the overall credibility of the findings.

Agreed. We have rewritten the statistical analyses and results sections extensively to attempt to incorporate these comments.

The revisions require a considerable revision

The article is resubmitted with considerable revisions which we believe improve the quality of the manuscript based on these comments – thank you!

Round 2

Reviewer 2 Report

Comments and Suggestions for Authors

The authors have revised the work and made changes. My feeling is that more changes would have enhanced the paper and they could have provided a stronger narrative to explain the study. That said, it gets through, but just enough.